# The Role of Innate Immune Response and Microbiome in Resilience of Dairy Cattle to Disease: The Mastitis Model

**DOI:** 10.3390/ani10081397

**Published:** 2020-08-11

**Authors:** Valerio Bronzo, Vincenzo Lopreiato, Federica Riva, Massimo Amadori, Giulio Curone, Maria Filippa Addis, Paola Cremonesi, Paolo Moroni, Erminio Trevisi, Bianca Castiglioni

**Affiliations:** 1Dipartimento di Medicina Veterinaria, Università degli Studi di Milano, 26900 Lodi, Italy; valerio.bronzo@unimi.it (V.B.); federica.riva@unimi.it (F.R.); giulio.curone@unimi.it (G.C.); filippa.addis@unimi.it (M.F.A.); pm389@cornell.edu (P.M.); 2Dipartimento di Scienze animali, Alimentazione e Nutrizione, Facoltà di Agraria, Scienze Alimentari e Ambientali, Università Cattolica del Sacro Cuore, 29122 Piacenza, Italy; vincenzo.lopreiato@unicatt.it (V.L.); erminio.trevisi@unicatt.it (E.T.); 3Rete Nazionale di Immunologia Veterinaria, 25125 Brescia, Italy; 4Institute of Biology and Biotechnology in Agriculture, National Research Council (CNR), 26900 Lodi, Italy; paola.cremonesi@ibba.cnr.it (P.C.); bianca.castiglioni@ibba.cnr.it (B.C.); 5Quality Milk Production Services, Animal Health Diagnostic Center, Cornell University, 240 Farrier Road, Ithaca, NY 14850, USA

**Keywords:** dairy cattle diseases, innate immune system, metabolic stress, microbiome

## Abstract

**Simple Summary:**

A major concern for the development of livestock activities is represented by the gradual reduction of antibiotic usage in farm animals, which may disturb the fragile balance between animal health and production. Therefore, it is necessary to maintain the immunocompetence of farm animals within the structure of this new trend toward reduced drug usage. High-yielding dairy cattle often experience more disease prevalence associated with short life expectancy and reduced environmental fitness. These signs of immunosuppression can be linked to metabolic changes observed around calving, which confirms the crucial link between immunity and milk production levels. The immunocompetence of these animals should be re-appraised and new disease control strategies should be based on creating a more efficient immune system. This review summarizes the dairy cow’s metabolic response to stress and what role the innate immune system and microbiome play. The review also discusses how new approaches to animal health based on specific intervention at dry-off and in the first weeks after calving are needed as the relevant stressors are pivotal to disease occurrence.

**Abstract:**

Animal health is affected by many factors such as metabolic stress, the immune system, and epidemiological features that interconnect. The immune system has evolved along with the phylogenetic evolution as a highly refined sensing and response system, poised to react against diverse infectious and non-infectious stressors for better survival and adaptation. It is now known that high genetic merit for milk yield is correlated with a defective control of the inflammatory response, underlying the occurrence of several production diseases. This is evident in the mastitis model where high-yielding dairy cows show high disease prevalence of the mammary gland with reduced effectiveness of the innate immune system and poor control over the inflammatory response to microbial agents. There is growing evidence of epigenetic effects on innate immunity genes underlying the response to common microbial agents. The aforementioned agents, along with other non-infectious stressors, can give rise to abnormal activation of the innate immune system, underlying serious disease conditions, and affecting milk yield. Furthermore, the microbiome also plays a role in shaping immune functions and disease resistance as a whole. Accordingly, proper modulation of the microbiome can be pivotal to successful disease control strategies. These strategies can benefit from a fundamental re-appraisal of native cattle breeds as models of disease resistance based on successful coping of both infectious and non-infectious stressors.

## 1. Introduction

In the last decade, ensuring animal health and welfare with the progressive reduction of drug usage has become a key issue for farmers as well as consumers worldwide. Dairy cattle diseases cause morbidity, mortality, and often decreased profitability for farmers, but antibiotics are now used more responsibly for treatment and control of these diseases [1,2]. Due to the known difficulties in developing novel antibiotic classes, the prudent use of the same products must be targeted. Public concerns have been raised regarding animal disease control, how animals for human consumption are treated with drugs, and the environment in which these animals are raised.

Alternative methods for preventing animal diseases are needed. One idea is through the modulation of the immune system. It has been documented that it is rare for every animal exposed to the same infection to develop symptoms that are clinical; furthermore, different breeds exhibit different traits related to disease [3,4,5,6]. It is difficult to explain why some animals in the same group develop varying degrees of the same illness. Genetics, the immune system, management, age, and other factors influence the health of an animal [7,8]. More variables play a role in animal health, making it difficult to pinpoint any single factor (Figure 1).

During the periparturient period, dairy cows undergo a number of metabolic-, endocrine-, physiologic-, and immune-related changes, rendering cows more susceptible to disease and less efficient. Health problems occurring before and after calving lead to severe negative effects on the productive efficiency of lactating cows. Reductions in the cow’s production and increased mortality rates are associated with periparturient health disorders. The costs of antimicrobial drugs, vaccines, labor, and preventive measures must be taken into consideration. During this period, immune system efficiency together with good liver functionality as well as the capability of cows to minimize the gap between nutrient intake (increasing dry matter intake )and nutrient output (milk production) determines the disease resistance capability of the animal [9]. The most important roles of the immune system are to prevent microbial diffusion and to reduce or eliminate infections.

## 2. Immunocompetence of High Genetic Merit Dairy Cattle and Disease Control Strategies

The immune system has developed along with the phylogenetic evolution as a refined sensing and response system, aimed at neutralizing all the possible noxa affecting or potentially affecting the host’s homeostasis [10]. The system has evolved from the recognition of conserved patterns of microbial pathogens to having great potential for recognizing fine specificities of microbial agents. Adaptive immunity rose with phylogenesis approximately 500 million years ago in jawed fish and proceeded to mammals as a result of selective pressures derived from the increased complexity of organs and apparata [11]. As a result, mammals avail themselves of innate immune mechanisms to deal with a plethora of infectious and non-infectious stressors. Adaptive mechanisms (antibodies and antigen-specific T lymphocytes) are used whenever the primary non-adaptive mechanisms fail to control a challenge to homeostasis [12]. 

Ruminants are no exception to this general rule. Domestication of ruminants began some 10,000 years ago [13] and has since played a vital part in the economic and social advancement of mankind. It can be argued that domestication was an advantage to ruminants in terms of easier access to feeding resources and protection against climatic challenges [14]. The relationship with mankind became complex with the advent of intensive farming and genetic selection for higher production levels. This relationship gave rise to a substantial worsening of animal welfare, and the historical relationship between domestication and welfare has become a bell-shaped dose-response curve [14]. We must find credible solutions to the major problem of ethics and the sustainability of farming activities. These solutions must take into consideration the diverging needs of environmental constraints and high production levels brought about by the increasing world population and its growing demand for animal products. 

In this conceptual framework, a major concern for the development of farming activities is represented by the gradual reduction of antibiotic usage in farm animals, which may disturb the fragile balance between animal health and production. It is necessary to stimulate the immunocompetence of farm animals within the structure of this new trend toward reduced drug usage. High-yielding dairy cattle often experience high disease prevalence associated with short life expectancy [15]. Most importantly, they show distinct signs of reduced environmental fitness, shown as coping poorly with both infectious and non-infectious stressors, as observed, e.g., in the hot summer season of 2003 [16]. The immunocompetence of these animals should be re-evaluated and new disease control strategies should be based on increasing the efficiency of the immune system.

### 2.1. The Concept of Immunocompetence

Immunocompetence is the ability of the body to produce a normal immune response following exposure to an antigen. This process involves complex genetic traits [17]. To produce an effective immune response, different cells and genes are necessary along with the ability of innate and adaptive immunities to coordinate. Danger describes the force that dictates the reaction profile of the immune system [18] for both infectious and non-infectious stressors [10]. Microbial infections entail some overlapping signals triggered by both PAMPs (pathogen-associated molecular patterns) and DAMPs (damage-associated molecular patterns). Within this operational framework, the innate immune system begins to destroy the stressors affecting the host’s homeostasis. Innate immunity must not cause substantial tissue damage as a result of a disproportionate inflammatory response. If pathogens persist after the innate response, adaptive immunity is induced to control the ongoing infection. The B and T cell receptor activity directed against specific antigens is the main component of immunocompetence. Innate immunity also plays an important part in the recruitment and orientation of receptor responses. These are used sparingly by the host as the response of secondary antibodies and immunological memory benefits represent high energetic cost [19]. Immunocompetence depends on factors such as a diet with adequate protein, energy, and multiple micronutrients. Immunocompetence presents sexual dimorphism where females present a general increased immunoreactivity compared to males [20]. Sexual dimorphism is due to genetic differences (several immune genes are in the X chromosome) and hormonal selective pressure. To achieve good reproductive fitness, females are selected to have a long life span due to a stronger immune system, whereas males need to maximize sexual mating early in life without investing in the immune system [21]. The immunocompetence of an individual undergoes some changes during the lifespan. Calves can adequately react to environmental pathogens through the transfer of colostrum immunoglobulins. Subsequent immunodeficiency or immune-compromised status in calfhood can occur following infections, drug treatments, and prolonged environmental stressful conditions. It is possible to enhance the immune response with three different general approaches: vaccination, passive immunization, and immunomodulation.

Evidence of reduced immunocompetence of high-yielding dairy cattle derives from epidemiological data and experimental studies [22,23,24]. As for the epidemiological data, the dramatic improvement of milk quality in terms of somatic cell counts was paired with an impressive increase in the milk yield of Holstein cows [25,26]. The impact of these performances on animal welfare and health has been considerable. In this respect, as the genetic ability to produce milk increases, more cows develop production diseases; the associations between increased milk production and increased risk of production diseases, as well as reduced fertility, are clearly documented, but less is known about the biological mechanisms behind these relationships [22]. Cows alive in the North-Eastern part of the USA at 48 months of age decreased from 80% in 1957 to 13% in 2002; on the same farms and in the same time period, the mean calving interval increased from 13 to 15.5 months [22]. As for experimental studies, high-yielding dairy cattle showed distinct signs of immunosuppression, which can be linked to the dramatic metabolic changes observed around calving [23,24]. Metabolic stresses associated with lactation influence the composition of peripheral blood mononuclear cell populations, as opposed to cows submitted to mastectomy [27], which confirms the crucial link between immunity and milk production levels. 

### 2.2. Metabolic Stress and the Innate Immune System

Innate and adaptive immune mechanisms are complementary and synergistic. This operational framework has been jeopardized by genetic selection for high milk yield, which led to reduced serum concentrations of lysozyme compared with the other cattle breeds [28]. Lysozyme plays a fundamental antimicrobial role and is part of important regulatory circuits of the inflammatory response [29]. Metabolic priority for offspring survival demands the maintenance of milk yield to the detriment of other functions [30] as the fetus and placenta have the same priority during pregnancy as the brain and Coagulase-negative Staphylococci (CNS) [12]. The high levels of milk yield exceed the potential of dry matter intake and the subsequent negative energy balance gives rise to metabolic stress, shown as a disequilibrium in the homeostasis of a living organism as a result of anomalous utilization of nutrients [31]. The unsatisfactory profiles of the immune response in high-yielding dairy cattle can be either primary (i.e., associated with the genetic selection for high milk/fat/protein yields), or secondary to metabolic stress (Figure 2). Two main signaling pathways monitor nutrient availability, control metabolic stress responses, and exert a central role in modulating innate immunity: the mTOR-(mammalian target of rapamycin) and eIF2α (eukaryotic initiation factor-2α)-dependent signal transduction cascades [31]. The most important regulators of mTOR and eIF2α are cellular energy status (ATP/AMP ratio), amino acid availability, oxygen tension, and oxidative stress. A direct link between metabolism and innate immunity is the binding of free fatty acids to Toll-like Receptors (TLR) 4, which is implicated in the development of inflammation in states of hyperlipidemia; the subsequent cascade of signaling events is very similar to that observed after exposure to microbial stressors [32].

Metabolic stress in the framework of Negative Energy Balance (NEB) should be seen as a crucial element underlying the occurrence of diverse disease conditions of dairy cows. In this respect, the roles of the p38a MAPK/mTOR and Pi3/Akt/mTOR signaling pathways are pivotal to regulating the balance between pro and anti-inflammatory cytokines in tissues in response to environmental stress [33]. This confirms the central role of the innate immune system in response to environmental stressors. Additionally, the potent regulatory actions of the immune system might be conveniently exploited in the future toward vaccines for metabolic diseases, like type 2 diabetes mellitus [34], which is reminiscent of insulin resistance in dairy cows.

### 2.3. The Influence of the Microbiome on the Immune System of Dairy Cattle

The microbiome contributes to the architecture and function of tissues, influences the host energy metabolism, and plays an important role in the balance between health and disease. The microbiota and its metabolites are crucial in the maintenance of host homeostasis [35]. At the beginning of the host’s life, the composition of the microbiota evolves into a healthy and viable community that strengthens itself and the host. Early development disturbances such as antibiotics, infections, or poor feed may lead to greater disease susceptibility [36].

Next-generation sequencing has enabled several groups to investigate microbiome influences on disease development. Bovine microbial communities have been described across many anatomical sites [37,38,39,40], including the mammary gland [41,42,43] and the uterus [44,45,46,47]. The composition of the bovine microbiome can affect the health [48,49,50] and performance of animals [51]. In ruminants such as dairy cows, the intestinal community of calves changes rapidly after birth and constantly during the first 12 weeks of life. Bacteroides–Prevotella and Clostridium coccoides–Eubacterium rectale species dominate the calf microbiota in this period [52]. After weaning, the microbiota must also compensate for the change in diet. This is a critical period when several events may affect the microbiota and health of the animal. During weaning Bacteroidetes decreased (remaining the dominant phyla) while Proteobacteria and Firmicutes increased [38]. These changes in microbial community composition, in part, are due to host physiological changes but are also likely due to the introduction of solid feeds because diet is a large driver of microbial community composition and modulation [38]. Weaning in dairy calves elicited an immune response in the lower gastrointestinal tract, but adding solid feed in addition to milk replacer resulted in changes to the immune response as well as gut bacteria [53]. Increased solid feed resulted in an increased total amount of bacteria present in the gastrointestinal tract. Weaning age and method of weaning can also change rumen or gastrointestinal tract microbiome establishment and community structure [54]. Weaning strategies can influence the ability of a calf to adapt to the dietary shift and can influence the severity of production losses. The effects of various feeding strategies and age as determining factors in the extent of microbial shifts in the rumen and feces during weaning still need to be studied.

Studies indicate that bovine gut microbiome can change immune responses [55]. These changes may be through direct and indirect mechanisms, such as through bacterial secretion of antimicrobial compounds or through influencing the expression of genes underlying host mucosal immune responses [56]. While much research on commensal microbes modulating host immune responses exists, studies investigating pathogen and pathobiont impact on host health are few [55,57]. The process of antimicrobial resistance is a relevant public health issue and, although antimicrobial use in human medicine arguably contributes to antimicrobial resistance much more than corresponding use in the livestock sector, it is important that farms proactively apply principles of prudent and judicious use of antimicrobials [58]. Farm animals and humans alike are threatened by this development. To promote early life wellness, pre- and probiotics are used to establish and restore microbiota health. Several studies demonstrated that probiotics and prebiotics achieved a positive balance in the gastrointestinal microbiota of cattle [59]. The functional interactions between gut microbes and relationships between microbes and host cells have yet to be fully investigated. Other pre-existent factors may play a crucial role, including host genetics, environmental conditions, and the resident, established microbiota [36].

## 3. Immunocompetence in Bovine Mastitis

Immunocompetence in the mammary gland is the outcome of a complex, coordinated network of anatomical, humoral, and cellular factors that are both specific and non-specific [60]. Immunocompetence can vary during lactation, showing depression in the peripartum period due to the hormonal and metabolic stress of calving and milk production [61]. The teat duct epithelium produces keratin that physically traps bacteria and blocks their migration to the mammary cistern. Keratin also has antimicrobial activity due to some bacteriostatic fatty acids (lauric, myristic, palmitoleic, and linoleic), as well as fibrous proteins that bind and damage the microorganism cell wall [62]. The duct epithelium can also modulate the expression of Pathogen Recognition Receptors (PRRs), such as Toll-like Receptors (TLRs), that after binding to PAMPs lead to the expression of inflammatory genes, and the release of cytokines (IL-1beta, TNF-alpha, IL-8) and acute phase proteins (haptoglobin, HP) but also antimicrobial peptides (pentraxin 3 PTX3, lipopolysaccharide-binding protein LBP) [6,63,64]. The mammary gland displays both innate and adaptive immune mechanisms that collaborate to defend the tissue against microbial invasions. The innate immune system is able to recognize the pathogens through TLRs and trigger an inflammatory response to kill them by phagocytosis (facilitated by the expression of surface receptors for Ig and complement proteins), or expression of antimicrobial molecules (lactoferrin, transferrin, lysozyme, defensins, cathelicidin, myeloperoxidases, complement system). Neutrophils respond with the up-regulation of adhesion molecules (L-selectin and beta2-integrin) to reach the damaged tissue [65]. Macrophages can perform phagocytosis, similar to neutrophils, but also act as antigen-presenting cells by exposing to lymphocytes the microbial antigens associated with Major Histocompatibility Complex MHC class II [66]. In the mammary gland of dairy cattle, the prevalence of lymphocyte populations (Th1, Th2, Th17, Treg, T cytotoxic, gamma-delta T cells, B cells) varies during lactation and the role of lymphocytes and natural killer cells is not fully understood under both health and disease conditions [67,68,69]. The main lymphocyte subset found in the bovine mammary gland is gamma-delta T cells [70]. During the peripartum period, lymphocytes assume a regulatory or suppressor phenotype, whereas at mid-lactation they shift to a cytotoxic phenotype and produce Interferon IFN-gamma [71]. In general, lymphocytes of the mammary gland are less responsive compared to the circulating ones; this could be partly due to a lower efficiency in antigen presentation in this area [61,67]. B cells do not change in number during the lactation stages [65]. B cells of the mammary gland serve as antigen-presenting cells but can also differentiate to plasma cells that produce antibodies (Ig) of four main isotypes: immunoglobulin G (IgG)1, IgG2, IgA, and IgM [72]. The concentrations of Ig in milk vary greatly during lactation. The activation of Th cells induces the expression of different cytokine repertoires that in turn influences the activity of T cytotoxic, B cells, macrophages, neutrophils, and Natural Killer NK cells [72]. The contribution to the immune response of mammary epithelial cells (MEC) is particularly important. MEC express PRRs; once they recognize microbial components, they can activate an innate immune response by expressing pro-inflammatory mediators (IL-1beta, TNF-alpha, IL-6, IL-8, acute phase proteins) and antimicrobial molecules (defensins, cathelicidin, calprotectin) [73]. Depending on the cytokine pattern in the inflamed mammary microenvironment, different responses can be started: Th1 (favoring cell-mediated responses), Th2 (favoring humoral responses), or Th17 (favoring activation and functions of granulocytes). This polarization drives the expression of specific molecules with different biological effects [74].

Decreased immunocompetence in the mammary gland, for instance in the peripartum period, predisposes cows to develop mastitis. In order to ameliorate the immune reactivity of the mammary gland, different strategies have been investigated. Diets can be fortified with minerals and micronutrients. These potentiate the activity of immune cells (such as Se and Vit E) and act as antioxidants (Vit A, Zn, Cu) to protect against the toxic effects of Reactive Oxygen Species (ROS) [75,76]. Another strategy consists in the administration of recombinant cytokines; for example, Granulocyte-Colony Stimulating Factor (G-CSF) results efficacious in the recruitment and differentiation from bone marrow reserves of a high number of polymorphonuclear leukocyte (PMN), as well as in the enhancement of their action at calving, which underlies a major reduction of mastitis prevalence [77]. Finally, vaccines were developed to prevent the insurgence of new infections and reduce the tissue damage induced by pathogens. The controversial results of different types of vaccines have cast doubts on this strategy [78]. It can be argued that merely using vaccines shows poor efficacy, which suggests their association with sanitary measures such as milking hygiene, teat dipping, confinement, and culling of chronically infected cows [79].

### 3.1. Epigenetics and Trained Immunity: Implications for the Control of Mastitis

Recently, evidence demonstrated that the innate immune system has the capability to develop “memory”, once attributed only to adaptive immunity. Innate immune memory is known as “trained immunity”. Studies show that the innate immune system can modify its response after the first encounter with both infectious and non-infectious stressors [80]. This is reminiscent of the cross-protection observed following different pathogen infections, described previously [81]. The peculiarities of trained immunity consist of the involvement of specific cell types (monocytes, macrophages, NK cells, innate lymphoid cells, ILC) and in epigenetic mechanisms that induce long-lasting adaptation. As a result, these cells remain highly responsive versus non-specific insults after the first recognition of a stressor [80]. In trained immunity, innate immune cells undergo epigenetic re-arrangement, leading to gene- or locus-specific changes in their chromatin profiles after a previous stimulation [80]. The major epigenetic mechanism active in trained immunity is histone modification with chromatin reconfiguration but other processes such as DNA methylation, modulation of microRNA, and long noncoding RNA expression seem to play a role [82]. This provides a transcriptional profile that modifies signaling and metabolism of innate immune cells [80]. Evidence of trained immunity is also described in dairy cows. Mammary epithelial cells stimulated with either Lipopolysaccharides (LPS) or Pam2CSK4 (two TLR ligands) develop endotoxin tolerance by epigenetic mechanisms; this response protects mammary tissues by enhancing the expression of beta-defensins and membrane protectors (Serum Amyloid A3 SAA3, Transglutaminase 3 TGM3) and down-regulating the expression of proinflammatory cytokines (TNF-alpha, IL-1beta) at a subsequent challenge [83]. This suggests that priming epithelial mammary cells with PAMPs induces a protective status by dampening exaggerated inflammation and enhancing bactericidal activity in a subsequent infection.

Trained immunity needs to be dissected to better understand its relevant subtended mechanisms. The identification of epigenetic changes increasing immunocompetence should be conducive to a promising mastitis control strategy. Furthermore, trained immunity in the mammary gland could be helpful in the design of efficacious vaccines combining both memory of adaptive immunity and “trained” innate responses.

### 3.2. The Milk Microbiome and the Mammary Gland Health

The bioactive molecules in milk play an integral role in training the immune system of recently born animals. The intestinal equilibrium of newborns is maintained by a synergistic relationship between antimicrobial peptides, lactoferrins, lysozymes immunoglobulins, and oligosaccharides [84,85]. The nutrient-rich ecosystem of milk allows growth of a wide range of microorganisms [41,86]. These microorganisms may contribute to the development of neonate gut microbiota, interact with the immune system, and regulate inflammatory responses and infection susceptibility [87].

Inflammation of the mammary gland, mastitis, is a response to intramammary infection, metabolic disorders, and trauma. Intramammary infection often occurs from the passage of pathogens beyond the teat canal [62], activating immune responses. Several factors can trigger mammary gland defenses against pathogens [88]. Commensal microbiota residing in the udder [41] may govern mastitis susceptibility. Bacteriocins produced by certain non-aureus Staphylococci (NAS) and *Corynebacterium* species colonizing the teat apices and teat canals may inhibit growth of major mastitis pathogens [89]. Within complex ecosystems, ecosystem diversity can increase resiliency against an influx of external species by supporting favorable interactivity [90]. The complexity of microbe to microbe communications concerning the functional properties of the mammary ecosystem are difficult to understand. It is essential to identify those bacterial species in the milk microbiota that contribute to mammary homeostasis and mastitis pathogen susceptibility [91]. While exploring milk microbiota diversity in relation to udder health, studies have shown a connection between dysbiotic microbiota and mastitis incidence [48,92,93,94]. In clinical *versus* non-clinical milk samples, clinical sample microbiota had reduced richness and evenness [48,93]. Despite these studies, much remains unknown concerning the ability of commensal microbiota to maintain mammary gland balance and modulate mastitis susceptibility. Derakhshani et al. [91] provided new insight into bacterial community composition and structure, which inhabit the mammary gland. This study shows the possible relationship of bacterial taxa with the inflammatory status of the udder. The identification of that possible hub species and candidate foundation taxa were associated with the inflammatory status of the mammary gland and/or future incidences of clinical mastitis. 

In conclusion, more research is necessary to understand the interactions between the microbial world and its hosts. The dissection of these relationships may result in new ways to repair microbial community structure in animals that are affected by organism imbalance.

## 4. Metabolic Response of Dairy Cows to Challenges: Insights into the Transition Period

It has been well established that high yielding dairy cows undergo several challenges throughout the whole gestation-lactation cycle with the most challenging time frame being in the transition period with the onset of lactation. Complex adaptation processes take place to enable the maintenance of the animals’ energy and nutrient homeostasis but many cows fail to successfully cope with the genetically imposed burden of meeting the requirements for the metabolically prioritized mammary gland in early lactation [28,95,96]. Metabolic stress produces a series of effects on productive and reproductive performance, on the immune system, and overall, on the well-being of the dairy cows [97]. Production diseases that imply a metabolic response are not necessarily related to performance level. Concerning the early lactation, Bertoni et al. [98] were able to develop a novel and clear interpretation of the relationship between milk production levels and health status. The authors demonstrated that high-yielding dairy cows with the highest milk production in the first month of lactation were characterized by better liver functionality and a less pronounced inflammatory status. Considerable biological variation in metabolic adaptation exists among individual animals, particularly in the period between late gestation and early lactation, which is accompanied by distinct levels of metabolic stress [99]. In this context, the successful adaptation to the lactational challenges relies on the metabolic robustness and the activation of the immune machinery [9], with performance levels that should not become another disturbing element of this adaptation. The period from late pregnancy over the course of calving is accompanied by a significant reduction in feed intake, causing the entrance into mild/severe NEB of dairy cows. The latter is the result of the sudden increase of milk secretion in early lactation, which is not compensated by a sufficient increase in feed intake, resulting in the metabolism and immunity (mainly the innate) being out of balance until the level of feed intake is able to cover the energy output with milk production.

Looking at later lactational stages, metabolic adaptation responses to environmental (heat stress, facilities, overcrowding, etc.) and immunometabolic stressors (acidosis, mastitis, etc.) have less detrimental effect on animal health because of the favorable energy balance and the unimpaired immune system. In this section, we presented an overview of the metabolic adaptations to different lactational stages, focusing on the transition period and on the effect of the negative energy balance (NEB) at mid and late lactation in comparison with the early lactation, all from a metabolic standpoint. 

### 4.1. A Multifaceted Challenge Called Transition Period

Drackley [95] argued that the biology underlying the transition to lactation was the “final frontier” in our understanding of the dairy cow. Since that time, a number of relevant in-depth studies uncovered most of the “obscured field” of the transition period with researchers demonstrating that immune cells are directly involved in a surprising array of metabolic functions including the maintenance of gastrointestinal function, control of adipose tissue lipolysis, which in turn determine liver functionality, and regulation of insulin sensitivity in multiple tissues [100,101,102,103]. It was also postulated and highlighted that metabolic changes related to energy and calcium supply in support of lactation occurring concurrently impair the innate immune response [9]. The NEB during the transition period explains this reduced immune function, which is also associated with increased concentrations of some blood metabolites as a result of tissue mobilization [96,104]. PMN and lymphocyte functions decrease gradually, starting about 2 weeks before calving, with the lowest efficiencies between the time of calving and 2 days after [104,105]. According to Kehrli et al. [104], the impaired neutrophil function during the periparturient period can be attributed to many of the hormonal and metabolic changes that prepare the mammary gland for lactation. Around this critical period, metabolism shifts from the demands of pregnancy to those of lactation, increasing demands for energy and protein. Together, these metabolic and immunologic challenges during the peripartal period are important factors that limit the ability of most cows to achieve optimal performance and immune-metabolic status [95,106]. Several “exploratory” studies on the immune function during the peripartum period led researchers to investigate potential interventions that might mitigate the immune dysfunction occurring immediately before and after parturition. The focus has been on stimulation of the circulating numbers and possibly the function of neutrophils using the recombinant bovine granulocyte colony-stimulating factor (rbG-CSF) as reported previously [58,107,108,109,110]. Treatment with rbG-CSF, starting from approximately a week before parturition with one injection (15 mg of rbG-CSF) and the second within 24 h after parturition, was able to increase neutrophils, basophils, eosinophils, and monocytes count [108,109,110,111]. From a molecular point of view, mRNA abundance of most genes involved in the cell adhesion, migration, recognition, antimicrobial activity, and inflammation cascade was increased. This suggested a complete activation of the immune machinery against the critical period post-partum, at least as a first response of leukocytes to transcriptional regulation [110]. Recently, Lopreiato et al. [102], for the first time, have highlighted the effect of rbG-CSF in maintaining stable cytokine levels during the first month after parturition, reflecting greater regulation of neutrophil recruitment, trafficking, and maturation during the inflammatory response, providing evidence of the immunomodulatory action of rbG-CSF around parturition, when dairy cows are highly immune hypo-reactive. A novel outcome reported by Lopreiato et al. [102] was that increasing the release of pro-inflammatory cytokines, interleukin-6 (IL-6) and interleukin-1β (IL-1β), after parturition upon rbG-CSF treatment did not result in increased systemic inflammation, as shown by haptoglobin and ceruloplasmin plasma levels. This latter finding points out that other mechanisms and/or molecules are likely to drive the inflammation after parturition. Plasma concentrations of IL-1β, IL-6, and tumor necrosis factor alpha (TNF-α) have been shown to be 1.5- to five-fold higher prepartum compared to the early lactation period [112]. Further studies should be undertaken to uncover the unknown mechanisms behind this controversial aspect of inflammation within the transition period. 

Besides a systemic inflammation, pro-inflammatory cytokines also act on peripheral cells inducing insulin resistance [113,114]. Under these conditions, circulating glucose is prioritized to the non-insulin-dependent glucose transporters that are expressed on immune cells and mammary glands only [115]. The massive glucose requirements of an activated immune system during systemic inflammation could further reduce the energy available for the other tissues, as the mammary gland does not markedly reduce the glucose uptake, aggravating the NEB occurring in early lactation [115]. When NEB occurs, mobilization of body fat and proteins is induced and free fatty acids (FFA) and amino acids are used as gluconeogenic sources by the liver [116]. A severe NEB occurring in the transition period could induce an FFA overload in the liver, inducing the release of beta-hydroxybutyrate (BHB) into the blood following ineffective oxidation of FFA and impaired pivotal functions [95,103,117]. This systemic inflammation is also known to induce the acute-phase response in liver, implying reduced constitutive protein expression (e.g., albumin, lipoproteins, paraoxonase, and retinol-binding protein), counterbalanced by augmented production of positive acute-phase proteins (APP) such as haptoglobin, ceruloplasmin, serum amyloid A, and C-reactive protein [118]. Oxidative stress also occurs during this period and is driven by the imbalance between the production of reactive oxygen metabolites (ROM), reactive nitrogen species (RNS), and the neutralizing capacity of antioxidant mechanisms in tissues and blood, caused by the increased immune response and the metabolic intensification to support lactogenesis [9,88]. When oxidative stress overwhelms cellular antioxidant capacity, ROM induces an inflammatory response. The increase in oxidative stress and inflammation during this period is also negatively associated with a reduction in liver functionality, and measurement of APP can provide a useful tool to assess liver function as well as inflammation [118]. Liver function is often impaired in transition dairy cows. In this context, it is relevant to point out the scenario occurring in the rumen during the transition period. Few studies have investigated the molecular adaptations of ruminal epithelium during the peripartum period [119,120,121,122,123]. These studies revealed the existence of interactions among genes of the immune system and those involved in the preparation for the onset of lactation [119,121,122], as well as the presence of growth factors that seem to be regulated after parturition [120]. The connections among ruminal fermentation, microbiota, the ensuing ruminal epithelium adaptations, and the consequence on systemic responses (e.g., immunometabolism) of the cow remain unclear. Whether microbial metabolism could affect epithelial gene expression via metabolites remains uncertain. The interaction of rumen content and epithelium with the systemic immune response opens a new scenario in the management of forestomaches. The role of saliva and its composition in terms of immune cells and immunogenic molecules should be further investigated as a potential factor of the reduced immunocompetence in dairy cows, mainly in the transition period [64,124]. Secondly, the role of diet appears crucial not only for an accurate formulation (e.g., fermentability of carbohydrates, protein degradability) but also for nutrient imbalance and/or for microbiota composition, which might alter epithelium functions (e.g., increasing its permeability). Several attempts to improve energy intake and thus avoiding detrimental effects especially after parturition have been proposed. Controlling energy intake during the dry period to near calculated requirements leads to transition success, with fewer diseases and disorders than cows fed high-energy diets [125,126,127], but also greater DMI around parturition [128]. Prolonged over-consumption of energy during the dry period can decrease post-calving DMI, resulting in negative responses of metabolism with higher NEFA and BHB in blood and greater triacylglycerol in the liver after calving [126]. The diet must be formulated to limit energy intake and meet the requirements for protein, minerals, and vitamins. To date, there little knowledge about diet formulation for the immediate postpartum period in order to optimize transition success and consequently reproduction efficiency. Proper dietary formulation in both dry and close-up periods would maintain or enable rumen adaptation to higher grain diets after calving, which in turn reflects a greater level of energy intake and energy utilization. 

Further perspectives are created by the opportunity to control rumen microbiomes by the host animal (genotype) as a result of genetic selection [129]. 

### 4.2. The Association between Rumen Microbiome, Cattle Production, and Health Traits

The rumen contains trillions of bacteria, protozoa, and methanogenic archaea as major components. A symbiotic relationship exists between a ruminant host and the microbiota where bacteria are provided shelter and nutrients and the host benefits from essential nutrients released by bacterial fermentation activities. Along with beneficial and essential nutrients, bacterial activity may be connected to the release of harmful compounds including bacterial endotoxins. 

Recent studies using an omics-based structure have suggested that differences in rumen microbiota are associated with cattle production and health traits, such as feed efficiency [130,131], methane (CH_4_) yield [132], milk composition [133], and ruminal acidosis [134]. Manipulating rumen microbiota may improve cattle productivity and health and reduce CH_4_ emissions. Transfaunation of ruminal contents is regularly used to enhance rumen function and milk production [135]. Although studies show that ciliated protozoa responsible for plant material digestion may be successfully transferred, there is more resistance within the bacterial community perhaps due to host-specific properties [136,137], suggesting the importance of the host’s genetics influence on rumen microbiota. More studies are needed to provide convincing information about associations between host and rumen microbiota.

## 5. Native Cattle Breeds, an Interesting “Case Study”

An interesting model for the study of the susceptibility to production diseases could be represented by native cattle breeds. These breeds have been part of livestock history until the 21st century when they were abandoned in favor of more productive cosmopolitan breeds (Holstein, Brown Swiss and Jersey) [138,139,140,141]. Some native breeds survived, thanks mainly to the efforts of many small traditional farmers residing mainly in rural marginal areas. The intense genetic selection received by the cosmopolitan breeds in order to improve the productive characteristics led them to develop peculiar physiological features, which have likely impaired some immune defense mechanisms, increasing the incidence of metabolic and infectious diseases, and worsening both fertility and longevity [76,142,143,144]. These phenomena have been studied in Holstein Friesian (HF) cows, the most widespread and highly selected dairy cattle breed. The intense genetic selection of HF for milk production has been associated with relevant physiological dysfunctions, e.g., reduction of the immune competence, severe NEB, inflammatory-like status, oxidative stress, and hypocalcemia [9]. The metabolic pressure caused by the high, energetic requests of the mammary gland combined with the stress resulting from the pregnancy-calving period in the context of severe NEB can lead to serious disruptions of physiological homeostatic balance [103,104]. All these physiological perturbations seem to be less intense in the “lower productive” native breeds. The scientific community is comparing breeds with different selective pressure in order to improve the comprehension of regulatory mechanisms of cattle physiology. The literature is not very extensive and has some limitations (e.g., number of animals, different environments, diet, and management), but important physiological information can be deduced. Mendoca et al. [145] compared HF and Montbéliarde-sired crossbred cows in the peripartum; no differences were found in terms of metabolic and inflammatory (haptoglobin concentration) responses, milk yield and incidence of typical peripartum diseases (retained fetal membranes, metritis, and subclinical endometritis). Despite that, the HF showed more pyrexia events in early lactation (50.0 vs. 31.4%) and a higher incidence of purulent vaginal discharge (44.2 vs. 26.5%) than crossbred cows. Curone et al. [6] compared HF with Rendena cows, an Italian breed native of the Rendena Valley in Northeastern Italy (Trentino), and observed that HF in the postpartum showed a more severe systemic inflammatory response in terms of haptoglobin, total proteins, globulins, and bilirubin, a more severe fat mobilization associated with lower body muscle mass and lower amino acid mobilization. In this study and in that of Cremonesi et al. [146], detailed insights into the milk microbial population of HF and Rendena along the transition period were also provided. The results highlighted the existence of differences in terms of general microbial diversity, taxonomy, and predicted functional profiles. Those differences might also have an impact on their mammary gland health concerning disease and pathogen resistance. These differences seem related to inflammo-metabolic changes occurring around calving, which suggest a possible relationship among these responses and the mechanisms of resistance in the mammary gland.

The local breed Simmental, when compared with HF cows during the transition period, presented a different metabolic adaptation, in terms of different energy, inflammatory, and oxidative pattern responses. Simmental showed a lower value of BHB and higher mobilization of muscle protein (creatinine) [101]. Simmental cows seemed more sensitive to induction of the immune system after calving, with a greater transcript abundance of proinflammatory cytokines and receptor genes, cell migration- and adhesion-related genes [102,110]. Begley et al. [147] showed that, when infected with Candida albicans, Norwegian Red cows have a greater primary antibody-mediated immune response, producing greater concentrations of immunoglobulin G (IgG) compared to HF cows. One of the largest comparative studies was performed by Bieber et al. [148]. They compared the production, fertility, longevity, and health-associated traits of local native and modern breeds of dairy cattle in 4 different European nations: Austria, Switzerland, Poland, and Sweden. They compared Original Braunvieh and Grey Cattle with Braunvieh (Brown Swiss blood >60%) in Switzerland; Grey Cattle with Braunvieh (Brown Swiss blood >50%) in Austria; Polish Black and White, Polish Red and White, and Polish Red with Polish Holstein Friesian in Poland; and Swedish Red with Swedish Holstein in Sweden. Average milk yields were substantially lower for local compared with commercial breeds in all countries. Local breeds showed a longer productive lifetime and a shorter calving interval with a lower insemination index than commercial breeds. Another approach to re-appraise the native breeds is the use of crossbreed cattle. Several studies showed how dairy producers may improve the longevity, robustness, and fertility of cows and the profitability of dairying by crossing pure HF cows with bulls of different native breeds [149,150,151]. The lower production level of local breeds is partly compensated by advantages in fertility, health status, and longevity. It is important to remember that the breeding goals should balance productivity with functional traits [152], and the choice of appropriate dairy breeds can be regarded as a key factor for successful health management in dairy farming.

## 6. Conclusions

The immune system has evolved along with the phylogenetic evolution as a highly refined sensing and response system poised to react against diverse infectious and non-infectious stressors for better survival and adaptation. This operational framework is jeopardized when high-yielding dairy cattle are poorly managed. Metabolic priority for offspring survival is affected by the levels of milk yield, exceeding the potential of dry matter intake. Secondly, the subsequent negative energy balance gives rise to metabolic stress, e.g., a disequilibrium in the homeostasis of a living organism as a result of anomalous utilization of nutrients. It can be argued that high genetic merit for milk yield is correlated with a defective control of the inflammatory response underlying the occurrence of several production diseases. This is evident in the mastitis model where high-yielding dairy cows show high disease prevalence in the framework of reduced effectiveness of the innate immune response.

Effective monitoring tools, immunomodulators, and nutraceuticals should be combined with proper farm management and feeding regimes. Specific intervention protocols should be implemented in the first weeks after calving and at dry-off because the relevant stressors are pivotal to disease occurrence and early culling of high-yielding dairy cattle.

## Figures and Tables

**Figure 1 animals-10-01397-f001:**
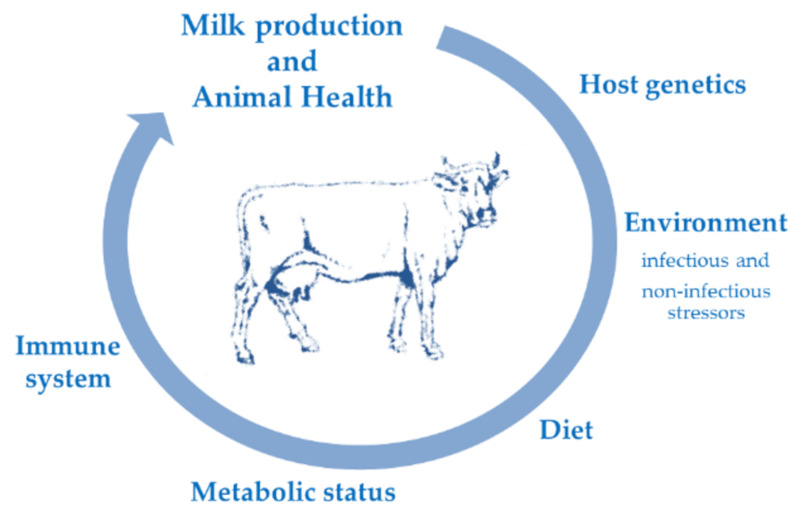
Milk production and animal health are influenced by and correlated to many factors such as genetics, environment stressors, diet, metabolic *status,* and the immunological system that all interact.

**Figure 2 animals-10-01397-f002:**
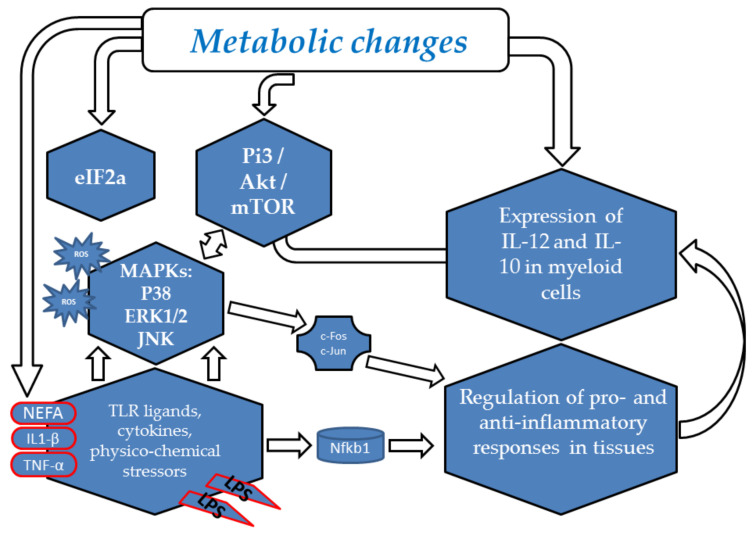
Metabolic stress is perceived by the innate immune system. The Pi3/Akt/mTOR- and eIF2α-dependent signal transduction cascades are the main signaling pathways monitoring nutrient availability and controlling metabolic stress responses. Toll-like receptors (TLRs) are expressed on innate immune cells, such as neutrophils, macrophages, and dendritic cells, and respond to the membrane components of Gram-positive or Gram-negative bacteria and to saturated free fatty acids mainly included in the NEFA (Non-esterified Fatty Acid) that are released by adipose tissue during status of negative energy balance. TLRs provoke rapid activation of innate immunity by inducing production of proinflammatory cytokines and upregulation of costimulatory molecules by both MAPK activation, which in turn activates c-Fos and c-Jun, and NF-kappa B activation through a MyD88-independent pathway. In addition, interleukin-1β (IL1-β) and tumor necrosis factor alpha (TNF-α), as critical cytokines, can induce a wide range of intracellular signal pathways as well as inflammation and immunity as nearly all cells express the respective receptors.

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
