# Peer review of "The Role of Innate Immune Response and Microbiome in Resilience of Dairy Cattle to Disease: The Mastitis Model"

_animals, 2020, doi:10.3390/ani10081397_

Round 1
Reviewer 1 Report
Overall, the manuscript by Broonzo and Lopreiato et al. is well written. A minor point: the title is inappropriate. What production diseases were reviewed in the manuscript? The authors mainly discussed about mastitis. How about other diseases? Why metritis was not reviewed? Besides infectious diseases, they would need to spend some efforts on metabolic disorders such as ruminal acidosis, ketosis, etc.
Author Response
Comments and Suggestions for Authors
Overall, the manuscript by Broonzo and Lopreiato et al. is well written. A minor point: the title is inappropriate. What production diseases were reviewed in the manuscript? The authors mainly discussed about mastitis. How about other diseases? Why metritis was not reviewed? Besides infectious diseases, they would need to spend some efforts on metabolic disorders such as ruminal acidosis, ketosis, etc.
Reply: The authors agree with the Reviewer that the immune system and the microbiome play a role both in many productive diseases and in metabolic disorders. However, to avoid the review becoming too long and difficult to read, the authors preferred to give a general overview, focusing only on mastitis as a model. Rather than introducing new chapters, the authors preferred to change the title, making it more consistent with the content. Therefore, the new title proposed is: The role of innate immune response and microbiome in resilience of dairy cattle to disease: the mastitis model.
Reviewer 2 Report
The manuscript by Bronzo et al., review the relationships between metabolism, the microbiome, and immunity on the health of early lactation dairy cows. The manuscript covers important information and is generally well-written. Some of the main concerns include the expression of quite pointed opinions (noted below), and the need for additional literature support. I would also like to see addition of discussion on the role of feed intake. Below are my specific comments:
Line 23: Please use a different word rather than implement
Line 25: more rather than high
Line 26: What is meant by ‘environmental fitness’?
Line 46: What is meant by ‘at large’?
Line 46-47: proper modulation of the microbiome can be
Line 54-55: Reference needed for this statement
Line 55-56: Please rewrite this sentence
Line 58-60: Please rewrite this sentence
Line 62: Raised rather than grown
Line 63: One idea is through the modulation of
Line 67-68: Reference needed for this statement
Line 68: Further, more variables
Line 70: Please revise this sentence. Cows are more vulnerable during the periparturient period, but it isn’t the period that makes them more vulnerable.
Line 72: Replace ‘we found’ with ‘are’
Line 74-75: It is more than just the immune system efficiency that allows an animal to resist disease. Metabolism/energy stores play a major role as well.
Line 103: Suggest replacing ‘meat’ with either ‘food’ or ‘animal products’
Line 108: What is meant by ‘Dairy cattle deserve scrutiny.’? Suggest removal
Line 110: What is meant by ‘environmental fitness’?
Line 124: Please don’t begin a sentence with an abbreviation. Here and throughout
Line 141-143: Please add a reference for this statement
Section 2.2. I would like to see expansion of this section. It seems the authors took a majority of the section to lead up to the implications of metabolic stress and implicating effects on mTor and eIF2a, but the discussion stops without any further discussion on the importance of these factors. Please expand.
Line 190: What is meant by ‘In beginning stages’? Is this beginning of life of the calf, beginning of the development of the microbiota? Please clarify.
Line 93: may lead to greater
Line 202: Please change the word ‘circumstances’
Line 202-204: What is the impact of these changes? Please elaborate.
Line 205: responses
Line 208: change ‘several’ to ‘much’
Line 210-211: Antimicrobial resistance is not a result of indiscrimate us by farm animals. Please revise sentence. Also, the cited reference [49] does not discuss antimicrobial resistance. This section appears very opinionated and it is suggested that the authors rewrite this section.
Line 212: I’m not aware of fecal transplants being used in livestock. Please add reference or revise sentence.
Line 215-216: There is substantial evidence on the role of probiotics on animal health and mode of action, particularly with yeast-based probiotics as well as lactobacilli species.
Line 216-219: Revise – run-on sentence.
Line 221-225: Please remove this section. Fecal transplant is not used in livestock so this section is irrelevant to the current topic.
Line 232: change ‘thanks to’ to ‘due to’
Line 257: change levels to concentrations
Line 270-271: rephrase sentence
Line 273-276: Please rewrite sentence
Line 277-278: Reference needed for this statement
Line 280: Please add citation to reference list
Line 303-307: Is it possible to ‘train’ the innate immune system utero to strengthen the immune system of dairy calves prior to birth?
Line 358-359: This sentence is unclear. Please revise.
Line 369, 376: change levels to concentrations
383-343: Please add reference
Line 415: While the role of the diet during this transition period is important, it must also be noted that there is a significant decrease in feed intake during this period. This not only impacts and potentiates the negative energy balance, but ultimately affects nutrient availability to lactation and the immune system. Perhaps so discussion on this should be added, in addition to discussion on the role of proper feeding of dairy cows prior to parturition in order to establish needed energy stores prior to parturition.
Line 426-428: Reference needed for statement
Line 430: Change ‘come up with’ to ‘develop’ or something similar
Line 510-514: No need to abbreviate cattle breeds as they not used more than once.
Line 528: Rephrase ‘huge levels’
Line 530: Remove ‘outright’
Conclusions: This section is long, repeats information from the Introduction, and therefore should be shortened.
Line 545: Remove ‘badly’
Author Response
Comments and Suggestions for Authors
The manuscript by Bronzo et al., review the relationships between metabolism, the microbiome, and immunity on the health of early lactation dairy cows. The manuscript covers important information and is generally well-written. Some of the main concerns include the expression of quite pointed opinions (noted below), and the need for additional literature support. I would also like to see addition of discussion on the role of feed intake. Below are my specific comments:
Reply: We thank the Reviewer for the comment.The authors added additional literature as well as additional discussion on the role of feed intake, as detailed below.
Line 23: Please use a different word rather than implement
Reply: the term “implement” has been replaced with “stimulate”
Line 25: more rather than high
Reply: the term “high” has been replaced with “more”
Line 26: What is meant by ‘environmental fitness’?
Reply: see definition hereunder for the comment to line 110
Line 46: What is meant by ‘at large’?
Reply: the term “at large” has been replaced by “as a whole” for clarity’s sake, at new line 47
Line 46-47: proper modulation of the microbiome can be
Reply: the sentence has been modified as suggested
Line 54-55: Reference needed for this statement
Reply: these two references have been added:
- Trevisi E, Zecconi A, Cogrossi S, Razzuoli E, Grossi P, Amadori M. Strategies for reduced antibiotic usage in dairy cattle farms. Res Vet Sci. 2014 Apr;96(2):229-33. doi: 10.1016/j.rvsc.2014.01.001
- Wemette M, Safi AG, Beauvais W, Ceres K, Shapiro M, Moroni P, Welcome FL, Ivanek R. New York State dairy farmers' perceptions of antibiotic use and resistance: A qualitative interview study. PLoS One. 2020 May 27;15(5):e0232937. doi: 10.1371/journal.pone.0232937. PMID: 32459799; PMCID: PMC7252592.
Line 55-56: Please rewrite this sentence
Reply: The sentence has been rewritten in: Dairy cattle diseases cause morbidity, mortality, and often decreased profitability for farmers but antibiotics are now used more responsibly for treatment and prevention.
Line 58-60: Please rewrite this sentence
Reply: This sentence has been deleted
Line 62: Raised rather than grown
Reply: the term “grown” has been replaced with “raised”
Line 63: One idea is through the modulation of
Reply: the sentence has been modified as suggested
Line 67-68: Reference needed for this statement
Reply: these two references have been added:
- Jeon SJ, Elzo M, DiLorenzo N, Lamb GC, Jeong KC. evaluation of animal genetic and physiological factors that affect the prevalence of Escherichia coli O157 in cattle. PLoS One. 2013;8:e55728.
- Bishop SC, Woolliams JA. Genomics and disease resistance studies in livestock. Livest Sci. 2014;166:190-198. doi:10.1016/j.livsci.2014.04.034
Line 68: Further, more variables
Reply: the sentence has been modified as suggested
Line 70: Please revise this sentence. Cows are more vulnerable during the periparturient period, but it isn’t the period that makes them more vulnerable.
Reply: We thank the Reviewer for the comment. The periparturient period encompasses several metabolic-, endocrine-, physiologic-, and immune-related changes (homeoretic adaptations) that in turn lead to a certain degree of vulnerability. The degree and length of time during which these systems remain out of balance could render cows more susceptible to disease, poor reproductive outcomes and less efficient. According to the Reviewer comment, the sentence was revised (new lines 67-69)
Line 72: Replace ‘we found’ with ‘are’
Reply: “we found” have been replaced with “are”
Line 74-75: It is more than just the immune system efficiency that allows an animal to resist disease. Metabolism/energy stores play a major role as well.
Reply: We thank the Reviewer for the suggestion. We revised the sentence accordingly (new lines 73-75)
Line 103: Suggest replacing ‘meat’ with either ‘food’ or ‘animal products’
Reply: the term “meat” has been replaced with “animal products”
Line 108: What is meant by ‘Dairy cattle deserve scrutiny.’? Suggest removal
Reply: The sentence has been removed
Line 110: What is meant by ‘environmental fitness’?
Reply: environmental fitness refers to the animals’ coping ability vis-à-vis a plethora of environmental stressors, including climate, nutrition, hierarchy position, as well as the infectious microbial stressors. This latter feature implies in turn major resilience and efficacy of the immune system. This concept is briefly addressed in the R1 version of the manuscript (new lines 109-110)
Line 124: Please don’t begin a sentence with an abbreviation. Here and throughout
Reply: the sentence has been modified as suggested
Line 141-143: Please add a reference for this statement
Reply: these two references have been added:
- Rupp, R., Boichard, D., 2000. Relationship of early first lactation somatic cell count with risk of subsequent first clinical mastitis. Livest. Prod. Sci. 62:169-180
- Hagnestam-Nielsen C, Emanuelson U., Berglund B, Strandberg- E. 2009. Relationship between somatic cell count and milk yield in different stages of lactation. J. Dairy Sci. 92 :3124–3133
Section 2.2. I would like to see expansion of this section. It seems the authors took a majority of the section to lead up to the implications of metabolic stress and implicating effects on mTor and eIF2a, but the discussion stops without any further discussion on the importance of these factors. Please expand.
Reply: We thank the Reviewer for the comment. Further details on the response to metabolic stress have been provided, adding these sentences at new lines 175-182: Metabolic stress in the framework of Negative Energy Balance (NEB) should be meant as a crucial element underlying the occurrence of diverse disease conditions of dairy cows. In this respect, the roles of the p38a MAPK/mTOR and Pi3/Akt/mTOR signalling pathways are pivotal to regulating the balance between pro and anti-inflammatory cytokines in tissues in response to environmental stress [33]. This confirm the central role of the innate immune system in the response to environmental stressors. In addition to that, the potent regulatory actions of the immune system might be conveniently exploited in the future towards vaccines for metabolic diseases, like Type 2 diabetes mellitus [34] which is reminiscent of insulin resistance in dairy cows..
Line 190: What is meant by ‘In beginning stages’? Is this beginning of life of the calf, beginning of the development of the microbiota? Please clarify.
Reply: the expression has been modified to avoid any misunderstanding.
Line 93: may lead to greater
Reply: the term “leave” has been replaced with “lead”
Line 202: Please change the word ‘circumstances’
Response: the term “circumstances” has been replaced with “events
Line 202-204: What is the impact of these changes? Please elaborate.
Reply: We thank the Reviewer for the comment. These new sentences have been added at new lines 213-224: These changes in microbial community composition, in part, are due to host physiological changes but also likely due to the introduction of solid feeds because diet is a large driver of microbial community composition and modulation [31]. Weaning in dairy calves elicited an immune response in the lower gastrointestinal tract, but adding solid feed in addition to milk replacer resulted in changes to the immune response as well as gut bacteria [32]. Increased solid feed resulted in increased total amount of bacteria present in the gastrointestinal tract [32]. Weaning age and method of weaning can also change rumen or gastrointestinal tract microbiome establishment and community structure [32]. Therefore, the weaning strategies can influence the ability of a calf to adapt to the dietary shift, and thus, can influence the severity of production losses. However, the effects of various feeding strategies during weaning as well as the weaning age as determining factor in the extent of microbial shifts in the rumen and faeces during weaning still have to be studied thoroughly.
Line 205: responses
Reply: the term has been modified
Line 208: change ‘several’ to ‘much’
Reply: the term “several” has been replaced with “much”
Line 210-211: Antimicrobial resistance is not a result of indiscrimate us by farm animals. Please revise sentence. Also, the cited reference [49] does not discuss antimicrobial resistance. This section appears very opinionated and it is suggested that the authors rewrite this section.
Reply: We thank the Reviewer for the comment. The sentence has been revised in: “The process of antimicrobial resistance is a relevant public health issue and, although antimicrobial use in human medicine arguably contributes to antimicrobial resistance much more as corresponding use in the livestock sector, it is important that farms proactively apply principles of prudent and judicious use of antimicrobials” (new lines 230-233)
In addition, the reference 49 has been changed with: Barkema HW, von Keyserlingk MA, Kastelic JP, Lam TJ, Luby C, Roy JP, LeBlanc SJ, Keefe GP, Kelton DF. Invited review: Changes in the dairy industry affecting dairy cattle health and welfare. J Dairy Sci. 2015 Nov;98(11):7426-45. doi: 10.3168/jds.2015-9377..
Line 212: I’m not aware of fecal transplants being used in livestock. Please add reference or revise sentence.
Reply: the sentence has been revised, removing fecal transplants.
Line 215-216: There is substantial evidence on the role of probiotics on animal health and mode of action, particularly with yeast-based probiotics as well as lactobacilli species.
Reply: The sentence has been modified, as showed below
Line 216-219: Revise – run-on sentence.
Response: The sentence has been revised in: “Several studies demonstrated that probiotics and prebiotics achieved a positive balance in the gastrointestinal microbiota of cattle [REF nuova However, the functional interactions between gut microbes, and relationships between microbes and host cells have yet to be fully investigated” (new lines 235-238)
The reference 51 has been replaced with; Uyeno Y, Shigemori S, Shimosato T. Effect of Probiotics/Prebiotics on Cattle Health and Productivity. Microbes Environ. 2015;30(2):126-132. doi:10.1264/jsme2.ME14176
Line 221-225: Please remove this section. Fecal transplant is not used in livestock so this section is irrelevant to the current topic.
Reply: The sentence has been removed.
Line 232: change ‘thanks to’ to ‘due to’
Reply: done
Line 257: change levels to concentrations
Reply: done
Line 270-271: rephrase sentence
Reply: First, diets can be fortified with minerals and micronutrients.
Line 273-276: Please rewrite sentence
Reply: The sentence has been re-written and further clarified.
Line 277-278: Reference needed for this statement
Reply: Scali F. et al., (2015) has been added as reference
Line 280: Please add citation to reference list
Reply: done
Line 303-307: Is it possible to ‘train’ the innate immune system utero to strengthen the immune system of dairy calves prior to birth?
Reply: We thank the Reviewer for the comment. To our knowledge, models of trained immunity have been only developed in animals after birth, as opposed to fetal life. Yet, we do not rule out such a possibility since epigenetic changes and trained immunity can be transmitted to the offspring (see report of S. Moorlag at International Veterinary Immunology Symposium, Seattle, USA, 2019, https://ivis2019.org/abstracts ). On the whole, we believe that it is still too early to speculate about this possibility.
Line 358-359: This sentence is unclear. Please revise.
Reply: We thank the Reviewer for the comment. The sentence has been revised at new lines 406-408
Line 369, 376: change levels to concentrations
Reply: done
383-343: Please add reference
Reply: Thanks. Reference has been added added at line 433
Line 415: While the role of the diet during this transition period is important, it must also be noted that there is a significant decrease in feed intake during this period. This not only impacts and potentiates the negative energy balance, but ultimately affects nutrient availability to lactation and the immune system. Perhaps so discussion on this should be added, in addition to discussion on the role of proper feeding of dairy cows prior to parturition in order to establish needed energy stores prior to parturition.
Reply: We thank the Reviewer for the suggestion. We added the discussion at new lines 374-379 and 464-479.
Line 426-428: Reference needed for statement
Reply: Reference has been added. L373-375
Line 430: Change ‘come up with’ to ‘develop’ or something similar
Reply: done
Line 510-514: No need to abbreviate cattle breeds as they not used more than once.
Reply: the abbreviations have been removed.
Line 528: Rephrase “huge levels”
Reply: “huge levels” gas been rephrased with the levels
Line 530: Remove ‘outright’
Reply: the term has been removed
Conclusions: This section is long, repeats information from the Introduction, and therefore should be shortened.
Reply: We thank the Reviewer for the suggestion. The conclusions have been shortened as suggested
Line 545: Remove ‘badly’
Reply: the term has been removed
Round 2
Reviewer 2 Report
The authors have addressed my concerns adequately.
Author Response
Reply: English has been revised as suggested, also with the engagement of Dr. Daryl Van Nydam and Belinda Gross, Cornell University, and Agnese Moroni, London School of Economics cited in the acknowledgments.